# The Impact of COVID-19-Related Living Restrictions on Eating Behaviours in Children and Adolescents: A Systematic Review

**DOI:** 10.3390/nu14173657

**Published:** 2022-09-04

**Authors:** Lucy Brakspear, Daniella Boules, Dasha Nicholls, Victoria Burmester

**Affiliations:** 1North East London Child and Adolescent Psychiatry Training Scheme, Great Ormond Street Hospital, Great Ormond Street, London WC1N 3JH, UK; 2Division of Psychiatry, 2nd Floor Commonwealth Building, Imperial College London, Du Cane Road, London W12 0NN, UK

**Keywords:** children, adolescents, young people, COVID-19, coronavirus, pandemic, living restrictions, lockdown, eating

## Abstract

The COVID-19 pandemic prompted the imposition of physical and social distancing measures worldwide. Emerging data suggest that younger age groups may be particularly vulnerable to the adverse mental health impacts of the pandemic. Since the start of the COVID-19 pandemic, there has been an unprecedented increase in demand for child and adolescent eating disorder services. The aim of this review was to systematically review and appraise the current literature on the impact of COVID-19-related living restrictions on the eating behaviours of children and adolescents. Searches of eight electronic databases were conducted in March 2021 and December 2021 for published and grey literature on eating behaviours of population samples of children and adolescents (aged 18 months to 18 years old) who were exposed to COVID-19-related living restrictions. Of 3165 retrieved references, sixteen studies were included in this review, comprising data from 125, 286 participants. There was a pattern towards healthier eating behaviours among children and adolescents during the COVID-19 lockdown. However, young people from lower socioeconomic groups showed a tendency towards more unhealthy eating behaviours, and there was an association between mood difficulties and greater changes in eating; this suggests that such groups may be more vulnerable to the adverse health consequences of lockdowns.

## 1. Introduction

COVID-19, an infectious disease caused by the SARS-CoV-2 virus, was declared a global pandemic by the World Health Organisation in March 2020 [1]; this prompted the imposition of physical and social distancing measures worldwide to limit the spread of the disease. Varying degrees of living restrictions, also referred to as quarantine or lockdowns, were ordered by governments on regional and national levels. Lockdowns were typically characterised by physical and social distancing between individuals, the closure of workplaces, shops, restaurants, and leisure facilities, home confinement, and various hygiene measures, such as hand washing and the wearing of facemasks. 

Data from previous epidemics, such as SARS, MERS, and Ebola, revealed a high prevalence of psychological stress and disorders including anxiety, depression, and post-traumatic stress disorder in the general population [2,3]. Data from the COVID-19 pandemic are revealing similar findings [4], with recent systematic reviews revealing that younger age groups may be particularly vulnerable to the adverse impacts of the pandemic on their mental health [5,6].

Eating behaviours of children are heavily influenced by family eating practices, including shared mealtimes [7], socioeconomic status [8], sociocultural norms [9], parental modelling of eating styles, and food choices [10]. The transition to adolescence typically sees peer influence [11] and the drive for independence also effect eating behaviour [12], with different patterns reported for males and females [13]. The factors influencing eating are likely to have changed significantly as a result of COVID-19 living restrictions.

With the closure of schools, home confinement, and reduced access to outdoor spaces, periods of lockdown resulted in significant disruption to young people’s daily lives. The resulting social isolation, uncertainty, COVID-19-related health anxiety, loss of freedom, and exposure to familial stress related to economic instability associated with the COVID-19 pandemic, particularly during periods of lockdown, has exposed young people to many known risk factors for mental illness [14]. Emerging studies on the child and adolescent population suggest that there are associations between the COVID-19 pandemic and negative mental health outcomes [15,16], with some studies noting increased rates of depression, anxiety [17,18], and eating disorders in this group [19].

There are reports that, over the course of the pandemic, there were increases in adult body weight [20] and paediatric obesity [21,22], which not only have well-established implications for general physical health [23,24] but are also associated with adverse outcomes related to COVID-19 infection [25]. In parallel, an unprecedented demand on child and adolescent eating disorder services has occurred, with significant increases in paediatric emergency department attendance related to eating disorders and their complications [26,27], inpatient admissions related to eating disorders, and exponential rises in referrals to children’s eating disorder teams [28].

Despite these increases in childhood obesity and referrals to eating disorders services during the COVID-19 pandemic, there remains little evidence-based understanding of the impact of the pandemic on the actual eating patterns of children and adolescents. A systematic review examining the changes in eating behaviours in this population during this period is crucial to enhance our understanding of the topic and inform vital service planning and provision, and government strategies and policies related to the funding of relevant and necessary mental health services.

Therefore, the aim of this systematic review was to provide a comprehensive exploration and critical analysis of the current data on population samples of children and adolescents exposed to COVID-19-related living restrictions, and to explore what changes in eating behaviours were observed. A secondary aim of the review was to determine, by examination of the context of findings, whether the data suggested any reasons for such changes.

## 2. Methods

The review was prospectively registered with the International Prospective Register of Systematic Reviews (PROSPERO, registration number CRD42021251515 [29]) and conducted in accordance with the Preferred Reporting Items for Systematic Reviews and Meta Analysis (PRISMA) guidelines [30] (see Appendix A for PRISMA checklist).

### 2.1. Search Strategy

Database searches of PubMed, PsycINFO, Embase, Web of Science, the WHO Global Research Database on COVID-19, and Google Scholar were conducted in March 2021 and repeated in December 2021. Grey literature was located within Google Scholar and the WHO research database, and additional searches for grey literature were conducted in paediatric collections within medRxiv and bioRxiv in April 2021. Academics with expertise in the field of eating behaviours were also consulted for their knowledge of studies of unpublished data on the topic.

Searches were conducted with keywords identified for each domain, as outlined in Box 1. Search terms were based on controlled vocabulary (MeSH terms), synonyms, related terms, and free terms, and combined with Boolean operators ‘OR’ for each concept and ‘AND’ within conjunctive expressions; truncations were applied using *.

Box 1Example search terms.COVID-19: Coronavirus OR Coronovirus OR COVID-19 COVID19 OR SARS-CoV-2 OR SARSCoV-2 OR SARSCoV2 OR SARS-CoV2 OR SARSCov19 OR SARS-Cov19 OR SARSCov-19 OR SARS-Cov-19.Eating behaviours: Eating OR eating behavio * OR eating habit * OR feeding OR diet * OR nutrition * OR weight OR eating disorder * OR disordered eating OR eating psychopathology OR emotional eating OR emotional overeating OR emotional undereating OR stress eating OR comfort eating OR Avoidant Restrictive Food Intake Disorder OR ARFID OR Restrictive Food Intake Disorder OR Anorexia Nervosa OR Bulimia Nervosa OR Binge Eating OR Purging OR Binge Eating Disorder OR Purging Disorder OR Other Specified Feeding OR Unspecified Feeding OR Restrictive Eating OR Other Specified Feeding and Eating Disorder OR restrained eating OR external eating behaviour OR food responsiveness OR food fussiness OR picky eating OR calorie counting.Age group: Child * OR adolescen * OR young people OR young person OR youth OR teen * OR kid * OR school * OR juvenile

### 2.2. Selection Criteria

Studies written in the English language that included data on changes in the eating behaviours of children and adolescents (from 18 months to 18 years of age) who were exposed to living restrictions during the COVID-19 pandemic were included. Peer-reviewed articles reporting primary data, including clinical and population surveys, along with cross-sectional, cohort, and longitudinal studies were included. Grey literature in the form of unpublished literature in the public domain, preprints, and studies in progress were also included.

Studies that included children and adolescents who may be on a specific diet due to a medical condition, who were enrolled in an obesity intervention or nutritional programme, and clinical populations of young people with eating disorders were excluded. Studies reporting solely on parental feeding practices rather than children and/or adolescent eating behaviours were also excluded. Ineligible study designs included review articles, opinion pieces, editorials without original data, case reports, conference abstracts, and letters. Studies reporting only qualitative data or unanalysed quantitative data were not included, nor were studies with an insufficient amount of data on eating behaviours or lack of change data.

### 2.3. Screening

Studies were imported from databases into COVIDence software [31] and duplicates were removed. Title and abstract screening was undertaken by two authors (LB and DB) independently to identify any articles that were irrelevant to the review. Full text reviews of all the remaining studies were then independently undertaken by the same two authors (LB and DB) to determine suitability for inclusion, according to the pre-specified inclusion and exclusion criteria. Any conflicts were resolved through review and discussion with a third reviewer (VB). 

### 2.4. Data Extraction

Data were extracted and tabulated into standardised tables developed for this review: the data comprised the author, year of publication, country, study design, sample size, participant characteristics, outcome measurement, main outcomes, other relevant findings, and study quality. Studies were categorised by study design, alphabetically. Where necessary, authors were contacted for additional information.

The primary outcome measurement was self or parent/carer reported changes in eating behaviour of children and adolescents. This included analysed frequency data on the consumption of certain food types which we categorised as ‘healthy’ or ‘unhealthy’, based on obviously healthy (e.g., fruit or vegetables) or unhealthy food types (e.g., sugary drinks or fried foods). Frequency data on snacking and meal frequency or patterns were also extracted. Where available, additional data on potential associations of eating behaviours with other variables (such as socioeconomic status) were reported in ‘other relevant findings’.

The strength of evidence was illustrated by denoting statistically significant results using * where *p* < 0.05 and ** where *p* < 0.001. Effect sizes were calculated for all statistically significant main outcomes, where means and standard deviations were available, and denoted as trivial (t), small (s), moderate (m), or large (l). It was not possible to pool data into a statistical meta-analysis due to the large degree of heterogeneity between study designs, samples, outcomes, and measures.

Data were prepared for synthesis by calculating effect sizes and reporting on the quality of each paper in the table to indicate the strength of the evidence. Data are presented by study design in the results table.

### 2.5. Study Quality and Risk of Bias

The methodological quality of studies was assessed using the Joanna Briggs Institute (JBI) Checklist for Analytical Cross-Sectional Studies [32], as all included studies used a cross-sectional study design. The JBI allows authors to critically assess the methodological quality of cross-sectional studies and determine the extent to which a study has addressed the possibility of bias in its design, conduct, and analysis, based on eight categories. Each study was independently assessed for quality by two authors (L.B. and D.B.), and any conflicts resolved following discussion and review with a third reviewer (V.B.), to determine if the study was of low, average, or high quality based on its performance across the categories.

Bias was limited by allocating two independent reviewers (L.B. and D.B.) to the screening process and the quality assessment for each included paper. Any disagreements were discussed with an independent third reviewer (V.B.).

## 3. Results

### 3.1. Identification and Selection of Studies

A total of 3165 references were retrieved in the database searches (1566 from the March 2021 database search, 1603 from December 2021). Five additional references were imported following contact with academics in the field. After the removal of duplicates, 2269 underwent title and abstract screening, of which 2101 were deemed irrelevant, leaving 168 full-text references to be assessed for eligibility. A total of 16 studies met the inclusion criteria and were included in this review (Figure 1).

### 3.2. Characteristics of the Included Studies

Study characteristics are summarised in Table 1. Dates of publication were all 2020 and 2021. The 16 included studies comprised a total of 125, 286 participants from 19 cohorts. The study populations included one with only children, six with just adolescents, and nine with both. Ten studies investigated Europe, two were in the Middle East, three in Asia, and one in Australasia. While all the studies were cross-sectional, only four collected data from the same cohort before and during lockdown, whereas three used different cohorts to obtain their data for the periods before and during lockdown. The remaining nine studies asked participants for their current (either during or after lockdown) eating behaviours, alongside retrospective estimates of pre-lockdown behaviours. Most studies collected data using bespoke questionnaires containing questions on food frequency. Five studies used recognised validated tools, including the Food Frequency Questionnaire (FFQ), a modified version of the FFQ (FFQ-6), the Binge Eating Scale, and the KIDMED (a tool used to evaluate adherence to the Mediterranean diet for children and youths). Nine studies used self-reported outcome measures, three used parent-reported outcomes, and four used a combination of parent- and self-reported outcomes. The majority of studies described changes in eating behaviours by comparing the consumption of certain food types via food frequency data before and during lockdown and after lockdown. Two studies reported on adherence to the Mediterranean diet as a proxy marker of healthy eating; one study reported on binge-eating symptoms only. Six studies included data on meal regularity, including snacking and breakfast consumption. Twelve studies were deemed to be of high quality, two average, and two low. The most commonly recognised limitations were the lack of description of specific lockdown conditions, and use of unvalidated measurement tools.

### 3.3. Changes in Eating Behaviours among Children and Adolescents Observed during the COVID-19 Lockdown

Of the 16 studies included, 15 reported changes in eating behaviours of the child and adolescent populations during lockdown; the effect sizes of the results ranged from trivial to large (see Table 1). One study (Muzi et al., 2021) reported no change in eating behaviours, but considered only binge-eating symptoms, rather than eating behaviours more generally. 

The main changes in the eating behaviours of children and adolescents observed during lockdown related to the frequency with which certain types of food were consumed. Seven studies described a pattern of increased healthy eating behaviours during lockdown by food type, whereas three reported a pattern of higher frequencies of unhealthy eating behaviours, and five found mixed results. 

Of the studies that reported a tendency towards healthy eating behaviours during lockdown, two referred to significantly increased adherence to the Mediterranean diet (Mastorci et al., 2021, *p* < 0.001 (s), Medrano et al., 2021 *p* < 0.05), two reported a significant increase in the consumption of markers of healthy eating (Kolata et al., 2021 (*p* < 0.001, Yu et al., 2021 (*p* < 0.05 (t)), one reported significant decreases in the consumption of markers of unhealthy eating (Munasinghe et al., 2020 *p* < 0.05 (m)), and one (Aguila-Martinez et al., 2021) reported significant increases in two markers of healthy eating (*p* < 0.05) alongside significant decreases in three markers of unhealthy eating (*p* < 0.05). Radwan et al., 2021 reported an overall trend towards healthier eating during lockdown according to an increase in the self-reported quality and healthiness of food (*p* < 0.001), a higher median ‘food quality score’ (*p* < 0.001), and increased ratings of eating healthy food as very good/excellent (*p* < 0.001), alongside increases in three markers of healthy eating (*p* < 0.001), and decreases in three markers of unhealthy eating (*p* < 0.001) in the presence of a decrease in one marker of healthy eating (*p* < 0.001). 

Three studies reported patterns towards more unhealthy eating behaviours during lockdown, but the quantity and quality of evidence for this finding was weaker. Al Hourani et al., 2021 reported significant increases in the consumption of seven markers of unhealthy eating (*p* < 0.05 − 0.001, (s,t)) vs. three markers of healthy eating (*p* < 0.05 − 0.001 (s)). Horikawa et al., 2021 reported a significant decrease in two markers of healthy eating (*p* < 0.001) and a significant decrease in the prevalence of ‘well balanced dietary intake’ during lockdown (*p* < 0.001). Segre et al., 2021 reported non-significant increases in three markers of unhealthy eating. 

The remaining five studies presented mixed findings. Androustous et al., 2021 reported significant increases in two markers of healthy eating (*p* < 0.001 (t,s)) and a significant decrease in one marker of healthy eating (*p* < 0.001 (t)), alongside increases in three markers of unhealthy eating, one of which was significant (*p* < 0.001 (m)). James et al., 2021 reported a decrease in two markers of unhealthy eating, of which one was significant (*p* < 0.05), alongside a significant increase in one marker of unhealthy eating (*p* < 0.05) and a non-significant decrease in one marker of healthy eating. Konstantinou et al., 2021 reported a significant increase in one marker of healthy eating (*p* < 0.001), alongside a significant increase in one marker of unhealthy eating (*p* < 0.001) and a non-significant decrease in one marker of unhealthy eating. Luszczki et al., 2021 reported a decrease in four markers of healthy eating, two of which were significant (*p* < 0.05 (t) and *p* < 0.001 (s)), alongside a decrease in two markers of unhealthy eating, of which one was significant (*p* < 0.001 (s)). Kim et al., 2021 reported significant decreases in three markers of unhealthy eating (*p* < 0.001), alongside a significant decrease in one marker of healthy eating (*p* < 0.001). 

With regard to the regularity of meals, one study (Aguila-Martinez et al., 2021) reported a significant decrease in the regularity of meals (*p* < 0.05). Two studies which reported significant decreases in snacking frequency (Luszczki et al., 2021, *p* < 0.05, Radwan et al., 2021, *p* < 0.001, respectively). Three studies reported significant increases in the frequency of breakfast consumption during lockdown (Kim et al., 2021, *p* < 0.001, Konstantinou et al., 2021, *p* < 0.05, James et al., 2021, *p* < 0.05).

The strength of the evidence presented and potential notable sources of bias must be acknowledged. The four studies with the most rigorous study design, involving a longitudinal follow-up of the same cohort before and during the pandemic, all reported trends towards more healthy eating behaviours (Aguilar-Martinez et al., 2021, Mastorci et al., 2021, Medrano et al., 2021, Munasinghe et al., 2020), and were assessed as high quality with a low risk of bias. All but one of the seven studies that reported a trend towards healthier eating behaviours were derived from self-reported outcomes, compared with other findings from parents or mixed methods of reporting. However, it must be noted that two of these studies report only on adherence to Mediterranean diet as a proxy indicator of healthy eating, without any further specific details of eating behaviours. 

The three studies that reported patterns of more unhealthy eating behaviours obtained results by asking participants to provide retrospective estimates of pre-lockdown eating behaviours, which imposes a risk of recall bias. The same three studies all used a combination of parent-reporting (for children) and self-reporting (from adolescents), which added to the risk of reporting bias as parents may not have been able to recall their child’s eating behaviours sufficiently accurately. The study conducted by Al Hourani et al., 2021 was deemed to have a high risk of bias, as the data collected during lockdown data were obtained during Ramadan, which would of course have been a period of atypical eating behaviours. Of the five studies that reported mixed trends in terms of healthy vs. unhealthy eating behaviours, three obtained data before and during the pandemic from different cohorts (James et al., 2021, Kim et al., 2021, Luszczki et al., 2021), two of which (James et al., 2021 and Kim et al., 2021) reported no significant differences in the participants’ baseline characteristics. The quality of the evidence and reliability of the results were particularly strong for Kim et al.’s 2021 study, due to the large sample size of 105,600 participants. The 2021 study of Luszczki et al. was, however, deemed to be a lower quality study as they did not describe the sociodemographic characteristics of each group, or state how they ensured that the cohorts were equally representative. The remaining two studies that reported mixed findings (Androustous et al., 2021, Konstantinou et al., 2021) were deemed to be high quality cross-sectional studies, although they did use a combination of self- and parent-reported outcomes, as did Luszczki et al., 2021.

### 3.4. Potential Explanations for Eating Changes

Other relevant findings were extracted from 11 studies, which described changes in eating behaviours in the context of other variables including age, sex, socioeconomic status, and others (summarised below). 

#### 3.4.1. Age

Segre et al., 2021 found that changed eating behaviours were most significant in primary school children (*p* < 0.001) compared with middle school children. Radwan et al., 2021 found that students aged 6–9 years had significantly higher food quality scores during the pandemic restrictions than before them (*p* < 0.001), whereas students aged 10–14 and 15–18 years had significantly lower scores during the COVID-19 restrictions (*p* < 0.001 for both). Students aged 6–9 and 15–18 years had significantly higher food quantity scores during the COVID-19 period (*p* < 0.001), but students aged 10–14 years had lower food quantity scores. 

#### 3.4.2. Sex

Aguila-Martinez et al., 2021 reported sex-based differences in the consumption of certain food types; the highest decrease in the intake of convenience food during lockdown was in girls (*p* < 0.05), whereas the consumption of sweets was the variable that decreased the most in boys (*p* < 0.05). Radwan et al., 2021 found that boys had significantly higher food quality scores during the pandemic (*p* < 0.001) compared with their pre-pandemic scores, whereas girls had higher pre-pandemic scores in food quality. 

#### 3.4.3. Socioeconomic Status

A number of studies reported trends towards increased unhealthy eating in lower socioeconomic groups. Aguila-Martinez et al., 2021 found a reduction in the consumption of fruits and vegetables, an increase in convenience food consumption, a decrease in the regularity of meals (*p* < 0.05), and an increase in meal skipping (*p* < 0.05). Such changes were higher among adolescents from a socioeconomic position perceived to be disadvantaged. Horikawa et al., 2021 reported that the lower the income group, the greater the rate of decrease in ‘well balanced dietary intake’ during lockdown (*p* < 0.001). James et al., 2021 found that children who had free school meals—a marker of socioeconomic deprivation—consumed significantly fewer fruits and vegetables during lockdown than those not receiving school meals. Radwan et al. reported that students from families with higher monthly incomes were found to have the highest quality (*p* < 0.001) and quantity (*p* < 0.001) food scores when compared to low and moderate-income groups, a disparity that increased significantly during lockdown (*p* < 0.001).

#### 3.4.4. Other Family Factors

Radwan et al., 2021 reported that students whose parents graduated from school or achieved at post-graduate level had a significantly higher food quality score during the COVID-19 period (*p* < 0.001). They also found that students from small families scored highest on food quality when compared to medium and large families, both before and during the COVID-19 pandemic (*p* < 0.001).

#### 3.4.5. Mood/Anxiety

Segre et al., 2021 found that a higher rather than lower frequency of mood symptoms was associated with changes in dietary habits (*p* < 0.05), but no significant association between anxiety levels and eating behaviours was found. 

#### 3.4.6. Social Media Use

Muzi et al., 2021 reported a correlation between increased problematic social media usage, with an increased total score of emotional-behavioural symptoms (*p* < 0.05), and increased binge-eating attitudes (*p* < 0.05) during lockdown. The role of problematic social media usage was explored as a potential predictor of more total and externalising problems and binge-eating attitudes, but no predictive model was statistically significant (*p* < 0.076).

#### 3.4.7. Shopping/Food Choice Patterns

Radwan et al., 2021 found that, during the pandemic, there was a decrease in the proportion of students whose families bought groceries every day (*p* < 0.001) and an increase in students who agreed or strongly agreed with the idea that, when purchasing food, they make choices according to calorie content and healthy properties (*p* < 0.001).

#### 3.4.8. Weight and BMI

Al Hoirani et al., 2021 reported an increase in BMI for age Z-score (*p* < 0.001) and a decrease in thinness and severe thinness (*p* < 0.001) during the pandemic. Kim et al., 2021 reported that subjective perceptions of body-shape image as obese were higher in the pandemic group than the pre-pandemic group (*p* < 0.001), and BMI was modestly, though significantly, higher in the pandemic group (BMI 21.3 vs. 21.5) (*p* < 0.001). Androustous et al., 2021 did not find statistically significant changes in weight, but their multiple regression analysis showed that body weight increase was associated with the increased consumption of breakfast, salty snacks, and total snacks, and with decreased physical activity. 

## 4. Discussion

Our findings were mixed, with seven studies showing a trend towards an increase in healthy eating by food type during the COVID-19 lockdown, three studies showing a trend towards increased unhealthy eating, and five presenting mixed findings. However, the strength of the evidence for increases in healthy eating behaviours by food type was greater, due to the high quality of these studies. Fewer studies with less rigorous designs presented evidence for unhealthy eating, including one study with a high risk of bias due to its collection of data during Ramadan. These findings are unexpected because eating disorder referrals [28] and rates of paediatric obesity [21] have increased during the pandemic (Al Hoirani et al., 2021, Kim et al., 2021). A potential explanation for our results is that our review did not include studies of clinical populations of obese children in intervention programmes, or eating disorder patients, whose eating behaviours would have more likely encompassed the extremes of the spectrum of eating behaviours. 

Existing literature from adult studies report similarly mixed findings. A number of adult studies show trends towards the increased consumption of markers of healthy eating; for example, a study of 10,116 Brazilian adults found a modest but statistically significant increase in the consumption of healthy eating markers and stability in the consumption of unhealthy food markers [49], findings which were replicated in other studies [50,51]. A 2021 study of 36,185 adults across 16 European countries found significant increased adherence to the Mediterranean diet during confinement [52]; similar findings were presented in another study [53]. Other studies show trends towards unhealthier eating patterns [54,55,56,57]. 

Our results also suggest that there was a decrease in snacking and an increase in breakfast consumption; this further supports our findings of a trend towards increased healthy eating behaviours during lockdown. Although these findings were statistically significant, the strength of these findings is limited by the small number of studies reporting on these outcomes, limiting our ability to draw reliable conclusions about such findings. In contrast to our findings, a number of adult studies [58,59], including a recent systematic review, have observed increases in snacking behaviours [60]. However, findings on breakfast consumption were more mixed in adult studies; one study of 164 young adult females found a significant increase in the number of women skipping breakfast during the pandemic [61].

Due to the limited data available and the lack of consistency between study findings and/or comparable data in the existing literature, it is difficult to reach any overall conclusions about the reported deterministic factors. However, our findings suggest that younger children’s eating behaviours changed more significantly during the pandemic than did those of adolescents, and that younger children tended to have an improved quality of food, whereas adolescents inclined towards reduced quality. However, the reliability and validity of these results are limited by the small number of studies that used age as a variable. A potential reason for these differences could be that, during lockdown, younger children consumed mainly parent-prepared food, whereas adolescents may have had more autonomy over their eating choices. 

Our findings suggest that children from lower socioeconomic groups displayed a trend towards more unhealthy eating patterns during lockdown, compared to those from less socioeconomically deprived backgrounds. This may be explained by a positive association between average food quality during lockdown and a higher level of parental education or smaller family size, which, as described by Radwan et al., 2021, is typically seen in families from higher socioeconomic backgrounds. Individuals from socioeconomically deprived backgrounds experienced increased food insecurity during lockdown [62,63]. This finding is also seen in adult studies, which report that food-insecure respondents were less likely than those who were food secure to eat healthy and nutritious food [64], specifically fruit and vegetables [65], and more likely to consume fast food and takeaways during lockdown [66]. 

Our findings of an association between mood and dietary habits (Segre et al., 2021) is supported by the existing literature. A study of 22,459 Chinese adults found that respondents who reported that the pandemic had had a higher psychological impact on them were more likely to increase their consumption of unhealthy foods [60]. Similarly, Burmester et al. [67] found that emotional eating increased due to the negative impacts of lockdown [68]. A 2021 systematic review concluded that COVID-19-pandemic-related uncertainty and anxiety contributed to an increase in subjective discomfort and raised the risk of compulsive eating as a strategy to cope with stress; a similar finding of an association between anxiety and increased hunger, emotional over-eating, and food and satiety responsiveness was found elsewhere [69]. An analysis of four UK longitudinal studies including 10,666 adults confirmed the relationship between poor mental health and adverse health behaviours, including diet, and found that associations were larger during lockdown compared with the pre-pandemic period [70]. 

Muzi et al., 2021 reported a correlation between more problematic social media usage with a higher total score of emotional-behavioural symptoms and more binge-eating attitudes during lockdown. The association between social media use and body dissatisfaction, the drive for thinness, and eating disorder vulnerability is well established [71,72,73]. COVID-19 pandemic studies suggest an increase in the use of social media during lockdown [74]. A study of 2601 Spanish women aged 14–35 years found statistically significant increases in the frequency of social media usage during lockdown and, specifically, in the relationships between the frequency of Instagram use and body dissatisfaction, the drive for thinness, and low self-esteem in the younger age group (14–24) [75]. These findings suggest that lockdown had an impact on social media use, and this might be linked to an increased drive for thinness and eating disorder risk, particularly among adolescents and young women.

At the time it was conducted, this systematic review was the first of its kind on the specific topic and population of interest, and is highly relevant to the clinical implications of the COVID-19 pandemic and to researchers. The review was conducted in accordance with PRISMA guidelines, and its risk of bias was reduced by use of two independent assessors and a third senior reviewer. Other strengths of this review include its incorporation of data from over 120,000 participants from various countries and continents worldwide, making its findings globally representative. The majority of the findings presented in this review were statistically significant, rendering them unlikely to have arisen due to chance. Effect sizes varied across results and studies (see Table 1), which should be taken into account when considering the clinical implications of the results. The effect sizes provided in this review, however, were derived from means that would not be expected to vary greatly, such as portions per day; in addition, the units of measurement varied greatly from consumption frequency per day vs. per week. It was not possible to conduct a meta-analysis of the results due to the heterogeneity of the outcome measurements. 

With regard to the limitations of the evidence included in the review, the cross-sectional design of the included studies means that the causal relationship between the COVID-19 pandemic and eating behaviours cannot be established in these studies; rather, associations can be drawn. Findings from all the studies were limited to comparing pre-lockdown and during lockdown, or immediately post-lockdown, without any further longitudinal data that might help us to understand whether any observed changes were longstanding. Studies that used retrospective estimates of pre-lockdown eating behaviours posed a risk of recall bias, potentially reducing the validity of these results. The results could also have been influenced by self- vs. parent-reported collection methods; however, the use of parent-reported results was necessary for data collection from younger age ranges. Additionally, it is important to be aware of the potential differences in eating behaviours of children and adolescents, due to the increased likelihood of parents preparing food for children, whereas adolescents may have been more likely to prepare food for themselves. Many studies did not use validated questionnaires to collect eating behaviour data; however, it was possible to obtain food frequency data from the bespoke questionnaires.

## 5. Conclusions

This review shows a trend towards healthier eating behaviours among children and adolescents during the COVID-19 lockdown, suggesting that being at home with parents might have a potential protective effect on children’s eating habits, particularly those of younger children. However, those from lower socioeconomic groups showed a trend towards more unhealthy eating behaviours during lockdown; moreover, young people with mood difficulties may be more vulnerable to eating changes, suggesting that such groups may be more vulnerable to the adverse consequences of lockdown and should be monitored specifically for this within paediatric and CAMHS services. Future research could compare clinical eating disorder and non-clinical populations, use validated eating questionnaires, assess the impact of social media on eating behaviour, and pay particular attention to longitudinal studies to examine the future impacts of the COVID-19 pandemic.

## Figures and Tables

**Figure 1 nutrients-14-03657-f001:**
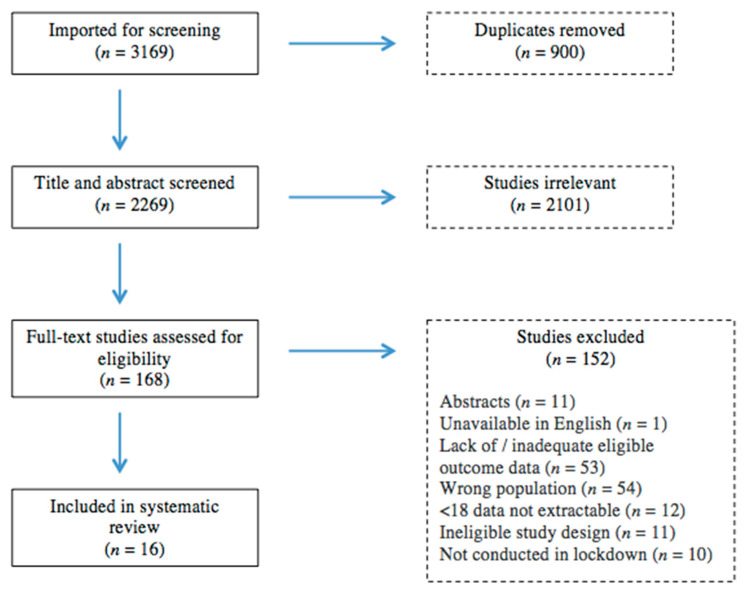
PRISMA flow chart.

**Table 1 nutrients-14-03657-t001:** Study characteristics.

Author, Year, Country	Study Design	Measurement Tool	Participant Characteristics	Main Findings	Additional Relevant Findings	Study Quality
Aguilar-Martinez et al., 2021 [33], Spain	Cross-sectional (pre-lockdown and during lockdown, 1 cohort)	Food Frequency Questionnaire (FFQ) (self-reported; online)	303 high-school students, 14–18 years, 70% female	2 × healthy marker increase (fruit * and vegetable * consumption). 3 × unhealthy marker decrease (soft drinks *, sweets and pastries *, and convenience foods). Decrease in regularity of meals *. Increase in snacking between meals. *Unit of measurement: increase/decrease/no change pre* vs. *during pandemic*	Reduction in fruit and vegetable consumption, increase in convenience food consumption, decrease in regularly of meals *, and increase in skipping meals * were significantly higher among adolescents from a socioeconomic position perceived to be more disadvantaged *. The highest decrease in the intake of convenience foods was in girls *; the consumption of sweets was the variable that decreased the most in boys *.	High
Mastorci et al., 2021 [34], Italy	Cross-sectional (pre-lockdown and during lockdown, 1 cohort)	KIDMED (self-reported; online)	1289 school children and adolescents (aged 10–14 yrs), 52% female	1 × healthy marker increase (adherence to the Mediterranean diet ** (Cohen’s d = −0.157 (s)). *Unit of measurement: KIDMED score*		High
Medrano et al., 2021 [35], Spain	Cross-sectional (pre-lockdown and during lockdown, 1 cohort)	KIDMED (self-reported; online)	113 school children and adolescents (aged 8–16 yrs), 48% female	1 × healthy marker increase (adherence to Mediterranean diet *). *Unit of measurement: KIDMED score*		High
Munasinghe et al., 2020 [36], Australia	Cross-sectional (1 cohort followed up weekly over 22 weeks, a period spanning before and during lockdown)	Bespoke questionnaire including food frequency data (online, self-reported)	582 adolescents (aged 13–19 yrs) from the general population, 61% females	1 × unhealthy marker decrease (fast food consumption *, Cohen’s d = 0.350 (m)). *Unit of measurement: servings per day*		High
James et al., 2021 [37], Wales	Cross-sectional (pre-lockdown and during lockdown, separate cohorts)	Bespoke questionnaire including food frequency data (self-reported, online)	2218 (1150 pre-pandemic; 1068 pandemic) school children (aged 8–11 yrs), 51% females	2 × unhealthy marker decrease (takeaways *, fizzy drink consumption). 1 × unhealthy marker increase (sugary snacks *). 1 × healthy eating marker decrease (fruit/veg consumption). Increase in frequency of breakfast consumption *. *Unit of measurement: frequency of consumption per week*	Those who had free school means had significantly lower fruit and veg consumption * and lower sugary snack consumption.	High
Kim et al., 2021 [38], Korea	Cross sectional (pre-lockdown and during lockdown, separate cohorts)	Bespoke questionnaire including food frequency data (self-reported, online)	105,600 middle- and high-school adolescents (53,461 pre-pandemic; 52,139 pandemic), aged 12–18 yrs, 52% female	1 × healthy marker decrease (fruit **) 3 × unhealthy marker decrease (fast food **, soda **, sweet drinks **). Higher frequency of eating breakfast **. *Unit of measurement: amount per week*	Reporting subjective body shape image as obese was lower in the 2019 group than in the 2020 group **. BMI was slightly though significantly higher in the 2020 group (21.3 vs. 21.5) **	High
Luszczki et al., 2021 [39], Poland	Cross-sectional (pre-lockdown and during lockdown, separate cohorts)	Modified Food Frequency Questionnaire (FFQ-6) (parent reported for children, self-reported for adolescents; online)	1017 (376 pre-pandemic group; 641 during lockdown group) school children (ages 6–12 yrs) and adolescents (aged 13–15 yrs), 51% females	4 × healthy marker decrease (legumes ** (Cohen’s d = 0.291 (s)), fish * (Cohen’s d = 0.081 (t)), raw veg, fresh fruits). 2 × unhealthy marker decrease (carbonated sugar-sweetened drinks, fast foods ** (Cohen’s d = 0.262 (s)). Decreased snacking * *Unit of measurement: frequency of consumption per day or week*		Average
Al Hoirani et al., 2021 [40], Jordan	Cross-sectional (current (during lockdown) and retrospective (pre-lockdown) estimates)	Food Frequency Questionnaire (parent reported for children, self-reported for adolescents; online)	447 children (6–12 yrs, 51%) and adolescents (13–17 yrs, 49%) from the general population, 52% female	3 × healthy marker increase (cooked veg for children * (Cohen’s d= −0.08 (t)) and adolescents ** (Cohen’s d = −0.14 (s)), raw veg for children * (Cohen’s d = −0.14 (s)) and adolescents * (Cohen’s d = −0.11 (s)), total fruits for children ** (Cohen’s d = −0.13 (s)) and adolescents ** (Cohen’s d = −0.16 (s)). 7 × unhealthy markers increase (carbonated drinks for children ** (Cohen’s d = −0.15 (s)) and adolescents * (Cohen’s d −0.13 (s)), fries for children * (Cohen’s d = −0.14 (s)) and adolescents ** (Cohen’s d = −0.19(s)), pizza * (Cohen’s d = -0.1(s)) and potato chips ** (Cohen’s d = −0.21(s)) for adolescents, sugar for children * (Cohen’s d = −0.1(s) and adolescents ** (Cohen’s d = −0.19 (s), ice cream for children ** (Cohen’s d= −0.2 (s)) and adolescents ** (Cohen’s d = −0.18 (s)), cake for children ** (Cohen’s d = −0.18 (s), trend in cake for adolescents. *Unit of measurement: servings per day*	Increase in BMI for age Z-score **, decrease in thinness and severe thinness **	Low
Androutsos et al., 2021 [41], Greece	Cross-sectional (current (during lockdown) and retrospective (pre-lockdown) estimates)	Bespoke questionnaire including food frequency data (parent-reported; online)	397 adolescents (12–18 yrs) from the general population, 49% females	2 × healthy marker increase (fruit ** Cohen’s d = −0.215 (s), vegetables ** Cohen’s d = −0.093 (t)). 3 × unhealthy marker increase (prepacked juices and sodas, salty snacks, sweets ** Cohen’s d = −0.333 (m)). 1 × unhealthy marker decrease (fast food ** (Cohen’s d = 0.045 (t)). Increased snack frequency ** (Cohen’s d = −0.597 (l)). Increased breakfast consumption frequency ** (Cohen’s d = −0.267 (s)). *Unit of measurement: servings per day*	Multiple regression analysis showed that body weight increase was associated with increased consumption of breakfast, salty snacks, and total snacks, and with decreased physical activity.	High
Horikawa et al., 2021 [42], Japan	Cross-sectional (current (during lockdown) and retrospective (pre-lockdown) estimates)	Bespoke questionnaire including food frequency data (parent-reported; paper postal questionnaire)	1111 children and adolescents (10–14 yrs) from the general population, 51% females	2 × healthy marker decrease (fruit ** and veg **). Decreased population of ‘well-balanced dietary intake’ during lockdown **. *Unit of measurement: consumption at least twice per day over one month.*	The lower the income group, the greater the rate of decrease in ‘well-balanced dietary intake’ during lockdown. **	High
Kolata et al., 2021 [43], Poland	Cross-sectional (current (during lockdown) and retrospective (pre-lockdown) estimates)	Bespoke questionnaire including food frequency data (self-reported; online)	1334 school children and adolescents (10–16 yrs), 53% females	2 × healthy marker increase (fruit ** and veg **). No significant change in markers of unhealthy eating (fast food, fried potato consumption). *Unit of measurement: portions per day.*		High
Konstantinou et al., 2021 [44], Cyprus	Cross-sectional (current (during lockdown) and retrospective (pre-lockdown) estimates)	Bespoke food frequency questionnaire (parent-reported; online)	1509 school children and adolescents (aged 5–14 yrs), 48% females	1 × healthy marker increase (consumption of fish **). 1 × unhealthy marker increase (consumption of sugary foods **). 1 × unhealthy marker decrease (consumption of ready-made foods). No change for fruit, vegetable, or legume consumption. Increased daily consumption of breakfast *. *Unit of measurement: number of days consumed per week.*		High
Muzi et al., 2021 [45], Italy	Cross sectional (one point during lockdown, compared with eating data from one pre-lockdown study)	Binge Eating Scale (self-reported; online)	62 adolescents aged 12–17 yrs from the general population; 63% females	No difference in binge eating scale outcomes. *Unit of measurement: Binge Eating Scale score*	Pandemic adolescents exhibited more problematic social media usage than their pre-pandemic peers (*p* < 0.001); more problematic social media usage was correlated with a higher total score of emotional-behavioural symptoms (*p* < 0.05) (*p* < 0.05) and more binge-eating attitudes (*p* < 0.05) (*p* < 0.05). The role of problematic social media usage was explored as a potential predictor of more total and externalizing problems and binge-eating attitudes, but no predictive model was statistically significant, all *p* < 0.076	Average
Radwan et al., 2021 [46], Palestine	Cross-sectional (current (during lockdown) and retrospective (pre-lockdown) estimates)	Bespoke questionnaire including food frequency data (parent-reported for <11 yrs, self-reported for 12 + yrs; online)	6398 school children and adolescents (aged 6–18 yrs, 88% 10–14 yrs), 80% females	5 × healthy marker increase (overall healthy food rating **, higher median ‘food quality score’ **, increase in students’ rating of their healthy food consumption as very good/excellent ** increased fruit consumption 2+/day **, more home-cooked meals **, and decrease in 1 other (lower veg consumption 2+/day **). 3 × unhealthy marker decrease (decreased fast food consumption **, decreased consumption of sodas/sweet tea 3+/day **, decreased desserts/sweets 4+ times/week **). Decreased snacking **. Decrease in food quantity ** *Units of measurement: number of days consumed on per week*	Boys had higher food quality scores during the pandemic **, whereas girls had higher pre-pandemic scores in food quality. Students aged 6–9 years exhibited a higher food quality score during the COVID-19 period than before the COVID-19 period **, whereas students aged 10–14 ** and 15–18 ** years attained a lower score during the COVID-19 period. Students aged 6–9 and 15–18 years had higher food quantity scores during the COVID-19 period **, but students aged 10–14 years had a lower score. Decrease in the proportion of students whose families bought groceries every day **. Increase in students who agreed or strongly agreed with the idea of choosing food according to calorie content and healthy properties when they buy **. Marked increase in students reporting fears about food hygiene from outside during the pandemic **.	High
Segre et al., 2021 [47], Italy	Cross-sectional (one time point during data, change data)	Bespoke (self- and parent-reported, video interview)	82 school children and adolescents (aged 6–14 yrs), 54.9 % primary school, 45.1 % middle school, 46% females	A significant proportion of children reported a perceived change in their eating habits *. Non-significant trend towards children reporting eating more during the pandemic, with a non-significant increase in the consumption of junk food, snacks, and sweets. *Units of measurement: increase/decrease /no change pre* vs. *during pandemic*	‘Changed eating behaviours’ were more significant in primary school children * compared with middle school children. Anxiety levels were not found to be significantly associated with changes in eating behaviours. Higher frequency of mood symptoms was associated with changes in dietary habits *.	High
Yu et al., 2021 [48], China	Cross-sectional (current (after lockdown) and retrospective (pre-lockdown) estimates)	Bespoke questionnaire including food frequency data (self-reported; online)	2824 high school students, 72% females	3 × healthy eating marker increase (fish * Cohen’s d = −0.077 (t), fresh veg * Cohen’s d = −0.045 (t), preserved veg * Cohen’s d = −0.056 (t)). *Units of measurement: frequency of consumption per week*		Low

* *p* < 0.05, ** *p* < 0.01. (t) = trivial effect size, (s) = small effect size, (m) = moderate effect size, (l) = large effect size.

## Data Availability

Not applicable.

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
