# Peer review of "The Impact of COVID-19-Related Living Restrictions on Eating Behaviours in Children and Adolescents: A Systematic Review"

_nutrients, 2022, doi:10.3390/nu14173657_

Round 1

Reviewer 1 Report

Thank you for the opportunity to review this article. The topic addressed by the authors with the manuscript entitled "The impact of COVID-19 related living restrictions on children and adolescent eating; systematic review" is of great interest and topicality.

I read the article carefully and found it well written, understandable and scientifically well organized.

I only have one comment to the authors. The citation regarding the PRISMA should be changed to the following and the flowchart revised accordingly.

Page MJ, McKenzie JE, Bossuyt PM, et al. The PRISMA 2020 statement: an updated guideline for reporting systematic reviews. BMJ. 2021;372:n71. Published 2021 Mar 29. doi:10.1136/bmj.n71

Author Response

Thank you to the reviewers for their time in reviewing this article for Nutrients. We believe their feedback greatly enhances our review. 

The first reviewer requested an updated reference for PRISMA, which we have now done.

Reviewer 2 Report

Dear authors, congratulations on the work carried out.

The article entitled: "the impact of covid-19 related living restrictions on children and adolescent eating; systematic review" deals with such a relevant topic as the change of habits in a pandemic situation, especially eating disorders and in the child and adolescent population.

It presents a systematic review of the literature on eating behaviours, including 16 studies, showing a healthier eating pattern at the time of the pandemic, and pointing out that the economic factor is related to unhealthy habits.

The methodology is adequate to replicate the review by another author.

The table showing the main findings is very illuminating.

The discussion is extensive and debates the studies.

My suggestions for improvement are:

-The introduction, in my opinion is poor, being necessary to expand it with studies on eating habits in childhood and adolescence.

-At the end of the discussion I recommend that you incorporate:

a) what does this work mean for researchers who read this work, does it help us in any way, what does this work imply.

b) what practical implications it has for the target population of this review, for children and adolescents.

c) it would be interesting to add after reading the studies to make the systematic review, what future lines of research you can extract and that are necessary for the future.

Congratulations for the work done.

Best regards.

Author Response

(1) The introduction, in my opinion is poor, being necessary to expand it with studies on eating habits in childhood and adolescence.

We value this observation and have now added a paragraph on children’s eating to the introduction

(2) At the end of the discussion I recommend that you incorporate:

  1. a) what does this work mean for researchers who read this work, does it help us in any way, what does this work imply.

Thank you for this feedback, we have now specified how the findings are relevant to researchers not just clinicians in the paragraph at line 419 and in the future directions section incorporated more research areas to help researchers understand what is outstanding.

  1. b) what practical implications it has for the target population of this review, for children and adolescents.

We agree that this is an important aspect to highlight and in our conclusion we feel this is addressed by the following text: “This review shows a pattern towards healthier eating behaviours of children and adolescents during COVID-19 lockdown, suggesting a potential protective effect of being at home with parents on children’s eating habits, particularly younger children. However, those from lower socioeconomic groups showed a pattern towards more unhealthy eating behaviours during lockdown, and young people with mood difficulties may be more vulnerable to eating changes; suggesting such groups may be more vulnerable to the adverse consequences of lockdown and should be monitored specifically for this within paediatric and CAMHS services.”

  1. c) it would be interesting to add after reading the studies to make the systematic review, what future lines of research you can extract and that are necessary for the future.

Thank you for this observation, we have expanded our future directions section, which now reads: “Future research could compare clinical eating disorder and non-clinical populations, use validated eating questionnaires, the impact of social media on eating behaviour and pay particular attention to longitudinal studies to examine the future impacts of the COVID-19 pandemic.”